# A Near-Infrared Mechanically Switchable Elastomeric Film as a Dynamic Cell Culture Substrate

**DOI:** 10.3390/biomedicines11010030

**Published:** 2022-12-22

**Authors:** Giovanni Spiaggia, Patricia Taladriz-Blanco, Stefan Hengsberger, Dedy Septiadi, Christoph Geers, Aaron Lee, Barbara Rothen-Rutishauser, Alke Petri-Fink

**Affiliations:** 1Adolphe Merkle Institute, University of Fribourg, Chemin des Verdiers 4, 1700 Fribourg, Switzerland; 2International Iberian Nanotechnology Laboratory (INL), Water Quality Group, Av. Mestre José Veiga s/n, 4715-330 Braga, Portugal; 3School of Engineering and Architecture (HEIA-FR), HES-SO, University of Applied Science and Arts in Western Switzerland, Boulevard de Pérolles 80, 1700 Fribourg, Switzerland; 4Department of Chemistry, University of Fribourg, Chemin du Musée 9, 1700 Fribourg, Switzerland

**Keywords:** PDMS, gold nanorods, substrate stiffness, cellular behaviour

## Abstract

Commercial static cell culture substrates can usually not change their physical properties over time, resulting in a limited representation of the variation in biomechanical cues in vivo. To overcome this limitation, approaches incorporating gold nanoparticles to act as transducers to external stimuli have been employed. In this work, gold nanorods were embedded in an elastomeric matrix and used as photothermal transducers to fabricate biocompatible light-responsive substrates. The nanocomposite films analysed by lock-in thermography and nanoindentation show a homogeneous heat distribution and a greater stiffness when irradiated with NIR light. After irradiation, the initial stiffness values were recovered. In vitro experiments performed during NIR irradiation with NIH-3T3 fibroblasts demonstrated that these films were biocompatible and cells remained viable. Cells cultured on the light stiffened nanocomposite exhibited a greater proliferation rate and stronger focal adhesion clustering, indicating increased cell-surface binding strength.

## 1. Introduction

Adherent cell types, such as fibroblasts and keratinocytes, actively sense variations in mechanical cues in vivo, modifying their behaviour in terms of actin fibre remodelling, focal adhesion phenotypes, and matrix biosynthesis capacity [1,2]. In particular, mechanical stiffness plays a pivotal role in cellular adhesion strength, migration, spreading, gene expression, and development and disease progression [3,4,5,6,7].

Static substrate models have been widely used to investigate cell responses to changes in substrate stiffness, resulting in a limited reproduction of the variations in biomechanical cues that occur in vivo mainly because these substrates cannot alter their physical properties in situ [8]. This limitation can be overcome by using dynamic in vitro systems that respond to, e.g., changes in temperature or pH, exposure to light sources, or electric and magnetic fields [9,10,11].

Dynamic in vitro systems comprising gold nanorods (Au NRs) [12,13,14,15] have been employed as stimuli-responsive materials in the biomedical field due to the easily tuneable optical properties of the Au NRs [16,17] and their capacity to generate and dissipate heat upon irradiation [18,19]. A couple of examples include the patterned NIPAM/Au NRs thermal actuators designed by Chandorkar et al. [20] and Sutton et al. [21]. In both studies, the irradiation of the substrates induces changes in cell migration associated with a change in the substrate stiffness.

Like the abovementioned studies, our approach comprises the use of Au NRs as photothermal transducers. However, the substrate consists of a collagen-coated polydimethylsiloxane (PDMS) flat film. Numerous publications report using PDMS as a cell substrate to study cellular mechanoresponse mainly because it is a transparent, biocompatible, flexible, and easy to manipulate and pattern elastomer whose mechanical properties are easily tuneable [22,23]. PDMS is also an ideal candidate for actuator applications as PDMS responds to temperature changes [24,25,26].

To the best of our knowledge, this is the first time Au NRs are exploited as stiffener agents on a substrate with a Young’s Modulus close to the mechanical response of native cartilage tissue (~MPa) [27]. Upon NIR irradiation, the heat dissipated from the Au NRs’ surface to the PDMS matrix leads to a 2.4-fold stiffening effect due to the constrained thermal expansion of the films. This leads to a faster proliferation of NIH-3T3 fibroblasts and a greater amount of focal adhesions (FAs) at the periphery of their actin filaments compared to the non-irradiated films. No signs of cytotoxicity were observed after 24 and 48 h of NIR illumination, confirming the biocompatibility of the fabricated substrate. The fabricated substrates could be potentially used to better understand, e.g., osteoarthritis based on chondrocytes proliferation and differentiation in aged articular cartilage [28,29].

## 2. Materials and Methods

Sylgard 184 Silicone Elastomer base, poly(dimethylsiloxane) (PDMS, Sylgard 184) and curing agent were provided by Dow Europe (Switzerland). Hexadecyltrimethylammonium bromide (CTAB, ≥98%), gold (III) chloride trihydrate (HAuCl_4_ ∙ 3H_2_O, ≥99.9%), sodium borohydride (NaBH_4_, ≥98%), silver nitrate (AgNO_3_, ≥99.9%), hydrochloric acid (HCl, ACS reagent, 37%), L-ascorbic acid (AA, 99%), cytotoxicity detection kit for lactate dehydrogenase (Catalyst, Diaphorase/NAD^+^ mixture, and Dye Solution INT and sodium lactate), glutaraldehyde solution (GA, 50 wt.% in H_2_O), Triton X-100, Bovine Serum Albumin (BSA, ≥99.9%), (3-aminopropyl)triethoxysilane (APTES, 99%), Fluoromount™ Aqueous Mounting Medium, and paraformaldehyde (reagent grade, crystalline) were obtained from Sigma-Aldrich (Darmstadt, Germany). Gibco™ collagen type I extracted from rat tail with a concentration of 3 mg/mL (solubilised in 20 mM of acetic acid), Invitrogen™ rhodamine phalloidin, 4′,6-diamidino-2-phenylindole, dihydrochloride (DAPI), Invitrogen™ ReadyProbes™ Cell Viability Imaging Kit (Blue/Green), goat anti-rabbit IgG (H + L) secondary antibody, DyLight 488, and phosphate-buffered saline (PBS-1X) tablets were purchased from ThermoFisher Scientific (Waltham, MA, USA). Dulbecco’s Modified Eagle Medium (DMEM, ATCC^®^ 0-2002™), penicillin/streptomycin solution (ATCC^®^ 30-2300™), Calf Bovine Serum (CBS), Iron Fortified (ATCC^®^ 30-2030™) and L-Glutamine solution (200 mM, ATCC^®^ 30-2214™), mouse fibroblasts NIH/3T3 (ATCC^®^ CRL-1658™) were acquired from ATCC^®^ (Manassas, VA, USA). Mouse TGF-beta 1 (TGF-β1) DuoSet ELISA (DY1679) and Recombinant Mouse IFN-gamma (IFN-γ) Protein (485-MI) were purchased from R&D Systems, Inc. (Minneapolis, MN, USA). Mouse monoclonal Collagen I alpha 1 Antibody (COL-1) was obtained from Novus Biologicals (Centennial, CO, USA). Recombinant rabbit monoclonal Anti-Paxillin antibody ([Y113], ab32084) was acquired from Abcam (Cambridge, UK). Milli-Q (Merck, Darmstadt, Germany) water was used for all preparations.

### 2.1. Preparation and Characterisation of Gold Nanorods (Au NRs)

Au NRs were prepared by the seed-mediated growth method [16,30]. Au seeds were synthesised by adding a freshly prepared NaBH_4_ aqueous solution (10 mM, 0.3 mL) to a mixture of HAuCl_4_·3H_2_O (50 mM, 0.025 mL) and CTAB (0.1 M, 4.7 mL) previously warmed to 28 °C in a 250 mL glass flask for 15 min. The seeds formation was confirmed by the change in colour to light brown. The dispersion was left undisturbed for 1 h at 28 °C. The gold growth solution was prepared by adding HAuCl_4_·3H_2_O (0.1 mM, 1.11 mL), AgNO_3_ (10 mM, 1.3 mL), HCl (1 M, 1.92 mL), and AA (0.1 M, 0.8 mL) to a CTAB solution (0.1 M, 100 mL) and mixed by inversion after each addition. Finally, the as-prepared Au seeds (480 μL, 0.25 mM) were added to the growth solution and left undisturbed overnight at 28 °C for the Au NRs formation. Ultimately, Au NRs were cleaned twice by centrifugation (8000× *g* for 50 min). Au NRs were visualised by TEM, operating at 120 kV (FEI Tecnai Spirit Microscope, Thermo Fisher Scientific, Waltham, MA, USA) and equipped with a 2048 × 2048 Veleta CDD camera (Olympus, Japan). Briefly, 5 µL of diluted Au NRs, final concentration of 20 μg/mL in Milli-Q water, was drop cast onto 300 mesh carbon membrane-coated copper grids (Electron Microscopy Sciences, Hatfield, PA, USA) and left to dry at room temperature before TEM analysis. The average size of the Au NRs, size distribution and aspect ratio were manually calculated using Fiji (USA), *n* = 150. The UV−Vis spectrum of Au NRs was recorded using a JASCO V-670 spectrophotometer using a 10 mm path-length quartz suprasil cuvette (Hellma Analytics, Müllheim, Germany).

### 2.2. Preparation of PDMS and PDMS/Au NRs Films

To precisely tune the thickness of the resulting films, PDMS was spin-coated for 1 min at 1000, 1500, and 2000 revolutions per minute (rpm). Briefly, Sylgard 184 Silicone Elastomer base and its curing agent were mixed at a ratio of 9:1 for 3 min, followed by degassing for 15 min. Subsequently, the mixture was poured onto a circular glass coverslip of 12 mm diameter (Circular Cover Glass #1.5, Thermo Fisher Scientific, Waltham, USA) previously cleaned with 70 vol.% of ethanol. The mixture was then spin-coated (WS-650 Hzb Spin Coater, Laurell Technologies Corporation, Lansdale, PA, USA) and left to cure at 70 °C overnight.

Three dispersions of Au NRs with Au^0^ concentrations of 0.6 mM, 0.39 mM, and 0.19 mM (1.5 mL) were centrifuged at 10,000× *g* for 50 min to remove the supernatant. Then, the supernatant was removed, and the pellets containing the Au NRs were manually mixed with 1.44 g mixture of Sylgard 184 Silicone Elastomer base and its curing agent at a ratio of 9 to 1 using a disposable glass Pasteur pipette (VWR, Dietikon, Switzerland) for 3 min. Thus, leading to an Au^0^ concentration in the viscous liquid of 7.5 wt.%, 5 wt.%, and 2.5 wt.%, respectively. Afterwards, the three Au NR mixtures underwent the same degassing, spin coating, and curing process as described for the pure PDMS elastomeric films. For simplification, in the manuscript, the concentrations were described as 7.5 wt.%, 5 wt.%, and 2.5 wt.% Au NRs.

### 2.3. Fluorescence-Enhanced Dark Field Microscopy

PDMS and PDMS/Au NRs containing 7.5 wt.% of Au^0^ films were visualised using a 100× objective lens and a numerical aperture of 0.8 in a Cytoviva dual-mode fluorescence-enhanced dark field microscopy setup (Cytoviva Inc., Auburn, AL, USA).

### 2.4. Lock-In Thermography Analysis (LIT)

To evaluate the photothermal conversion and the macroscopic distribution of Au NRs embedded in films, experimental LIT measurements were carried out (*n* = 5) using a custom-made setup as previously reported in the literature [31]. Briefly, the heat generated upon irradiating the films with a homogeneous multi-wavelength LED-based light source (AN178_2 61 LED module, ADOM, Germany) centred at 525 nm (power density of 74.2 mW/cm^2^) was recorded with an infrared camera (Onca-MWIR-InSb-320, XenICs, Leuven, Belgium) mounted on a standard microscope stand (Leica Micro-systems, Wetzlar, Germany). A light-homogenising glass rod (N-BK7, Edmund Optics Inc., Barrington, NJ, USA) placed between the LED panel and the sample holder ensures a homogeneous illumination of the films. To convert the acquired amplitude signal into 2D heating maps (in Kelvin), a custom-made LabVIEW-based software was developed to demodulate the infrared images according to the digital lock-in principle. Analysis of the resulting amplitude images was performed with Fiji (USA) to extract signal mean and standard deviation values.

### 2.5. Scanning Electron Microscopy (SEM/EDX) Characterisation 

The thickness and the distribution of the Au NRs within the PDMS films were determined by SEM and energy-dispersive X-ray spectroscopy using a Tescan Mira3 LM FE (Tescan, Brno, Czech Republic), coupled with an EDX detector (Octane Pro, AMETEK Inc., Berwyn, PA, USA) on platinum-coated films. EDX signals were acquired by performing a line scan on the films at an accelerating voltage of 20 kV and were further analysed with a TEAM™ EDS Software Suite (AMETEK, Inc., Berwyn, PA, USA).

### 2.6. Film Surface Functionalisation by Collagen Type I and Characterisation

Briefly, films were plasma-activated in a controlled O_2_ rich environment for 1 min (Low-pressure Zepto plasma coater, Diener Electronics, Ebhausen, Germany) and silanised by immersion in a 10 vol.% APTES solution in ethanol for 2 h. The unbonded APTES was removed by rinsing the films with ethanol twice. The following layer was obtained by incubating the films with a 2.5 vol.% glutaraldehyde solution in water for 1 h, followed by 3 rinses with Milli-Q water. The collagen layer was grafted by immersing the films in a 20 μg/mL solution of collagen type I prepared in 20 mM acetic acid for 2 h. All the steps were carried out at room temperature. Finally, samples were repeatedly washed with PBS and kept in deionised water at 4 °C until further use.

A drop of 5 μL of Milli-Q water was placed on un-coated and coated films (*n* = 3) to ensure a uniform droplet size, and the contact angle formed at the interphase was determined by fitting a tangent to the drop profile using an OSC 15Pro goniometer (DataPhysics instruments, Filderstadt, Germany). Fourier transform infrared (FTIR) spectroscopy of un-coated and coated films was also performed using a Perkin Elmer (USA) Spectrum 65 FTIR spectrometer from 1800 and 1500 cm^−1^ with a resolution of 4 cm^−1^. Visually, the homogeneity of the coatings was evaluated by staining the collagen type I with a two-step immunohistochemistry. Briefly, a rabbit monoclonal anti-collagen I (dilution 1:100 in PBS) was used as the primary antibody and incubated overnight. On the following day, after 3 PBS rinses, goat anti-rabbit DY488 secondary antibody (dilution 1:100 in PBS) was added and incubated for 2 h before visualisation. Samples were then mounted on glass slides with a Fluoromount™ Aqueous Mounting Medium and excited with a 488 nm continuous laser at a magnification of 10×. The fluorescence was collected using a 488 emission filter at 1024 × 1024 pixels resolution. Dried surface samples were also visualised by SEM.

### 2.7. Mechanical Measurements

The mechanical properties of pure PDMS, non-irradiated and irradiated PDMS/Au NRs films were characterised in a CSM (Anton-Paar) Ultra Nanoindenter [32]. Films were irradiated with an 810 nm collimated LED using a bandpass filter centre in 790 ± 2 nm with an FWHM = 10 ± 2 nm (Thorlabs, Newton, NJ, USA) for 1 h to ensure optimal heat distribution. Immediately after irradiation, the indentations were performed with a Berkovich tip with Young’s Modulus equal to 1140 GPa (E_tip_) and a Poisson’s ratio of 0.07 (ν_tip_). Briefly, 20 indentations for the 2000 rpm fabricated samples and 10 indentations per sample for the 1000 and 1500 rpm fabricated films were analysed employing force curves consisting of three stages. Only 10 indentations were used for the 1000 and 1500 rpm fabricated films because they did not undergo a change in stiffness under irradiation. All indentations were run to a maximum force of 200 μN with a loading rate of 200 μN/minutes, followed by a 10 s holding period at maximum force. In the last stage, tip unloading at 200 μN/minutes was performed. This third part shows the elastic recovery of the material and is used to calculate the Reduced Modulus (E_r_) via
$$S\;=\frac{2}{\sqrt{\pi }}\times E_{r}\times \sqrt{A_{c}}$$
where S is the initial slope of the unloading curve and A_c_ is the contact area, a tip dependent shape parameter. The reduced modulus is then used to evaluate the elastic modulus of the sample E_sample_ via the following equation:$$\frac{1}{E_{r}}=\frac{1-\nu_{\mathrm{sample}}^{2}}{E_{\mathrm{sample}}}+\frac{1-\nu_{\mathrm{tip}}^{2}}{E_{\mathrm{tip}}}$$
using a Poisson’s ratio of the sample of $\nu_{\mathrm{sample}}=0.5,$ as previously reported in the literature for PDMS [33]. The elastic modulus and the Poisson’s ratio of the diamond tip are known parameters.

### 2.8. Fibroblasts (NIH/3T3) Culture on the Functionalised Elastomeric Films

Mouse fibroblasts NIH/3T3 were cultivated in cDMEM (DMEM supplemented with 10 vol.% CBS, 1 vol.% penicillin/streptomycin and 1 vol.% of L-Glutamine) using a T75 flask obtained from TPP (Switzerland) at 37 °C, with a relative humidity of 95% and 5% of CO_2_ until reaching 80% confluence. Cells were washed with 10 mL of PBS and trypsinised with 1.5 mL Trypsin for 5 min. Then, 3 mL of cDMEM was added, and cells were counted using an automatic cell counter (NanoEnTek, Waltham, MA, USA). Approximately 40,000 cells (360 cell/mm^2^) supplemented with 0.6 mL of cDMEM were seeded and incubated on PDMS, non-irradiated PDMS/AuNRs and irradiated PDMS/AuNRs films for 24 and 48 h. For the experiments under NIR irradiation, cells were seeded immediately before switching on the NIR light and the latter was kept throughout the cultivation, i.e., 24 and 48 h.

### 2.9. Analysis of Cell Viability by the Lactate Dehydrogenase (LDH) Assay

Cytotoxicity was assessed immediately after 24 and 48 h of NIR irradiation by measuring the release of lactate dehydrogenase into the supernatant due to cell membrane disruption using a commercial LDH detection kit (Roche Applied Science, Mannheim, Germany); 100 μL of NIH/3T3 mouse fibroblasts (1500 cells/mm^2^) cultured on the films (*n* = 3) and bare wells used as a positive control (100 µL of 0.2 vol.% Triton X-100 in PBS) (*n* = 3) were mixed with 100 µL of the LDH assay kit and placed in a 96-well plate the day before to run the LDH assay. The absorbance at 630 nm was recorded using Bio-Rad Plate Reader (Switzerland). The data of each measurement was normalised by the mean of the positive control values.

### 2.10. Growth Factor Assay 

TGF-β1 secreted by cells into the supernatants was quantified using the respective ELISA DuoSet Development Diagnostic Kit immediately after 24 and 48 h of NIR irradiation, respectively; 1 μg/mL of IFN-γ in cDMEM was used as a positive control.

### 2.11. Confocal Laser Scanning Microscopy (cLSM), Immunohistochemistry, and Cell Viability 

The biological response of cells cultured on the films was assessed by cLSM (Zeiss LSM 710 meta, Germany). Samples were washed 3 times with PBS immediately after 24 and 48 h of NIR irradiation, fixed with a 4 vol.% solution of paraformaldehyde for 15 min, and washed 3 additional times with PBS to undergo immunofluorescence staining. Next, films were rinsed in 0.1% Triton X-100 in PBS for 5 min, and blocked with a 1% BSA in PBS for 20 min. For paxillin visualisation, rabbit monoclonal anti-Paxillin antibody diluted in PBS (1:100) was incubated overnight at 4 °C, followed by the addition of goat anti-rabbit DY488 secondary antibody diluted in PBS (1:100) for 2 h. F-Actin cytoskeleton and the nuclei of the cells were stained with rhodamine phalloidin diluted in PBS (1:80) for 1 h and DAPI diluted in PBS (1:100) for 5 min, respectively. Finally, the samples were washed 3 times with PBS and stored in 1 mL of PBS at 4 °C for further analysis. The subsequent fluorescence data was collected using DAPI, rhodamine and Alexa Fluor™ 488 emission filters at a resolution of 1024 × 1024 pixels after sequential excitation using a 405, 488 and 540 nm continuous laser and a 25× objective. Cell viability was evaluated by employing a commercial LIVE/DEAD assay, Invitrogen™ ReadyProbes™ Cell Viability Imaging Kit (Blue/Green) following manufacturer’s protocol immediately after 24 and 48 h of NIR irradiation. The number of surviving cells was calculated by employing the following equation:(1)$$\mathrm{Survival}\;\left(\%\right)=\frac{\mathrm{Live}\;\mathrm{cells}}{\mathrm{Live}\;\mathrm{cells}+\mathrm{Dead}\;\mathrm{cells}\;}\times 100$$

### 2.12. Image Analysis

Image analysis of cLSM, SEM, and TEM data was performed using Fiji (NIH, USA). For the cytotoxicity assay, images were thresholded to discard bias signal followed by binarisation (Otsu’s method) and calculation of compromised and uncompromised nuclei.

### 2.13. Statistical Analysis

Two- or one-way analysis of variance (ANOVA) was used to analyse the data using GraphPad PRISM software (USA). Significance for all statistical analyses was defined as *p* < 0.05, and all values are reported as the mean ± standard deviation of the mean for the technical and independent biological triplicates.

## 3. Results and Discussion

### 3.1. NIR Light Responsiveness of Nanocomposite Films

Au NRs, with a plasmon band centred at 803 nm, i.e., within the first biological window (650–1000 nm) and an aspect ratio of 3.0 ± 0.8 (Figure 1B and Figure A1 in the Appendix A), were embedded in a PDMS matrix at a concentration of 7.5 wt.% with respect to the PDMS weight (hereafter, referred as to PDMS/Au NRs) and spin-coated at 2000 rotations per minute (rpm) (Figure 1A,C). After overnight curing at 70 °C, the resulting flat films showed a thickness of 26 ± 0.5 µm. Pure PDMS films were prepared under the same conditions and used as a control. When imaged using lock-in thermography (LIT) [34,35] the resulting heating maps were uniform, indicating a homogeneous distribution of Au NRs within the films. Au NRs embedded in the PDMS were visualised by dark-field microscopy (Cytoviva) due to their enhanced scattering properties (Figure 1D, Inset).

In the present study, the elastic modulus was extrapolated from indentation experiments, considering a uniform, equi-biaxial distribution of the thermal stresses within the PDMS/Au NRs films when irradiated with NIR (hereafter referred to as PDMS/Au NRs NIR). This approach is justified by the homogeneous distribution of heat observed in the thermographic images. Residual stresses which are generated by thermal expansion can strongly affect the mechanical properties of materials in the short and long term [36], and nanoindentation is reported to be a valuable local-probe method to characterise residual stresses within a sample by analysing the load–displacement curve, especially for high-strength materials with large yield strain [37]. Among the mechanical properties which can be evaluated through indentation, the elastic modulus is reported to be affected by the thermal stresses distributed within the sample [38].

As shown in Figure 1E, nanoindentation experiments carried out for pure PDMS and PDMS/Au NRs films show a Young’s Modulus of 10.9 ± 2.4 MPa for PDMS/Au NRs NIR, corresponding to a 2.6-fold and 2.4-fold increase compared to pure PDMS (4.1 ± 0.5 MPa) and non-irradiated PDMS/Au NRs films (4.6 ± 0.8 MPa), respectively. Interestingly, after switching off the irradiation and letting the films rest for 1 h (hereafter referred to as PDMS/Au NRs OFF), the stiffness values decreased to 4.5 ± 0.6 MPa, confirming the relaxation of the thermal stress. The excitation of the free electrons in the plasmon band upon irradiation with a NIR light source increases the Au NRs surface temperature via electron-electron scattering, which is then dissipated to the surroundings within minutes [18,39]. Consequently, the physical constraint generated between the glass and the films prevents their free dilatation, leading to the accumulation of internal compressive stresses and thus, a stiffening effect [38]. It is worth mentioning that considering the optical loss of PDMS (<0.5 dB/cm) in the NIR region, no change in stiffness is expected for pure PDMS films under NIR irradiation [40,41].

Lower concentrations of Au NRs (2.5 wt.% and 5 wt.%), as well as spinning speeds (1000 and 1500 rpm) were also investigated. Nonetheless, the stiffening effect was only achieved by the films loaded with 7.5 wt.% Au NRs prepared at 2000 rpm. Higher loadings were also attempted; however, Au NRs underwent aggregation. Therefore, these formulations were not further investigated. Independent of the concentration of Au NRs, thinner films were obtained when increasing the spinning speed from 1000 to 1500 and 2000 rpm. Subsequently, a lower amount of Au NRs per unit of area was estimated for the films prepared at 2000 rpm. The latter is consistent with the energy-dispersive X-ray spectroscopy (EDX) data, showing a gradual decrease in elemental gold content for higher spinning speeds, explaining the lowest temperature reached for the films prepared at the highest speed (Figure A2 and Figure A3 in the Appendix A). Thus, our results indicate that the reversible stiffening effect assessed for the PDMS/Au NRs films results from a combination of the film thickness and the concentration and distribution of the Au NRs within the film.

### 3.2. Stiffening Effect on Cellular Behaviour

To assess the biocompatibility of PDMS/Au NRs NIR and the cellular responses when cultured on the stiffened substrates, in vitro experiments using NIH-3T3 fibroblasts were performed at 24 and 48 h. In order to promote cell adhesion and facilitate biocompatibility, films were covalently coated with a layer of collagen type I [42,43]. The success of the coating was assessed by contact angle and Fourier transform infrared spectroscopy (FTIR) (Figure A4 in the Appendix A). It is worth mentioning that the collagen effect on the mechanical properties was neglected due to the softer nature of the coating and by performing deep indents (~3.2 µm). Fibroblasts cultured on pure PDMS and non-irradiated PDMS/Au NRs were used as a control.

The ratio of live/dead stained cells obtained after performing a commercial LIVE/DEAD assay at 24 and 48 h shows little to no cytotoxic effects (Figure 2). Similarly, the lactate dehydrogenase (LDH) assay confirmed no cytotoxicity when cells were cultured on the control films (PDMS/Au NRs) and PDMS/Au NRs NIR for 24 and 48 h.

cLSM was used to visualise cell morphology and FAs by staining F-actin and paxillin, respectively (see the Section 2 for further details regarding the immunohistochemistry) (Figure 3). FAs have a major role in delivering mechano-based information across the cell through the interaction of membrane-bound integrin receptors linked to focal adhesion kinase (FAK) and paxillin [44,45,46]. These transduced signals to the intracellular compartments have a drastic effect on cellular processes, including migration, proliferation and secretion profile [47].

Interestingly, at 24 h, fibroblasts cultured on the activated films had a larger amount of focal adhesions at the periphery of their F-actin filaments. This observation was more pronounced at 48 h, and it is consistent with several research outputs confirming a larger clustering of FAs at the interface between the cell and its surroundings when cultured on stiffer substrates (Figure A5 in the Appendix A) [48,49]. Moreover, a larger FA surface area is linked to a greater traction force exerted by the cell toward the substrate [44,50].

To evaluate the effect of higher binding strength on the cellular proliferation rate, cells were trypsinised and automatically counted after 24 and 48 h (Figure 3). At 24 h, a greater number of cells was found on PDMS/Au NRs NIR compared to pure and non-irradiated PDMS films. Moreover, at 48 h, a significantly higher amount of cells compared to 24 h was counted, confirming that the greater the traction force exerted by the cells, the higher the proliferation rate [51]. To further confirm greater FA clustering by fibroblasts when cultured on PDMS/Au NRs NIR films, levels of secretion of TGF-β1 by fibroblasts were quantified by employing an ELISA assay (see the Section 2 for more details). IFN-γ and glass were used as a positive and negative control, respectively, because IFN-γ is involved in the upregulation of TGF-β1 in dermal and corneal fibroblasts [52] and glass is commonly used for evaluating cellular secretion capabilities in comparison to polymer-coated samples [53].

Lower values of TGF-β1 were detected when cultured on the PDMS/Au NRs NIR films compared to non-irradiated and pristine PDMS substrates. This is in accordance with the literature, in which it has been reported that higher TGF-β1 values lead to a gradual decrease in clusters of FA at the end of the F-actin filaments [54,55].

The increase in stiffness on PDMS/Au NRs NIR seemed to have a slight effect on cell morphology (Figure 3 and Figure A5 in the Appendix A). However, within this stiffness range, due to the greater stiffness of the substrate compared to the cytoskeleton, cells are not able to deform the underlying substrate and stretch their cytoskeleton with the same magnitude of spreading observed on tissue culture plastic [6,7].

## 4. Conclusions

In this study, we fabricated a light-responsive elastomeric film able to be stiffened in situ when irradiated with NIR light. A 2.4-fold increase in elastic moduli was achieved by employing Au NRs as photothermal transducers. The ability to impact the nanocomposite film stiffness in situ promotes their usage as dynamic cell culture substrates. Fibroblast cultured on the irradiated films showed major FA clustering at the end of F-actin filaments and lower TGF-β1 secretion at 24 and 48 h. Fabricated substrates are biocompatible, and there is no effect on cell morphology. Based on our results, these novel photo-responsive films are promising candidates as substrates for altering cellular responses in situ compared to static cell culture conditions.

## Figures and Tables

**Figure 1 biomedicines-11-00030-f001:**
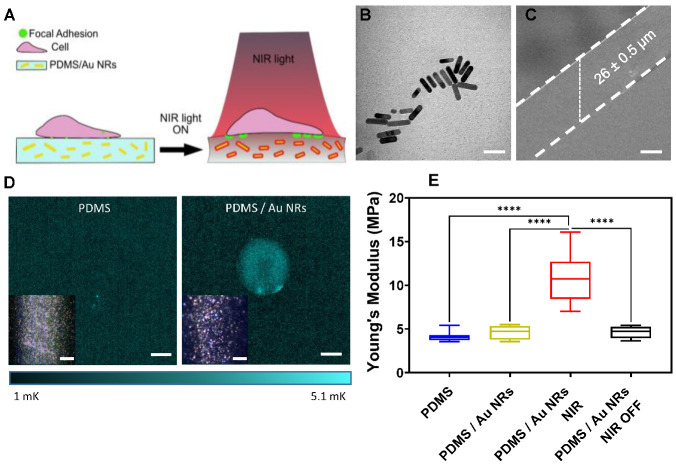
(**A**) Schematic representation of the light-responsive films used as cell culture substrates. Au NRs with LSPR centred at 803 nm were embedded in PDMS and used as photothermal transducers to manipulate cellular response in situ. By exposing the nanocomposite films to a NIR light source centred at 790 nm, a stiffening effect was achieved due to the constrained dilatation of the film, resulting in the clustering of FAs by fibroblasts. (**B**) TEM micrograph of the synthesised Au NRs. Scale bar: 50 nm. (**C**) SEM micrograph showing the lateral view of the films. Scale bar: 20 µm. (**D**) LIT thermographs obtained upon irradiation with a light-emitting diode (LED) source centred at 525 nm for pure PDMS and PDMS/Au NRs films. Scale bar: 5 mm. Inset Cytoviva images of pure and PDMS/Au NRs films. Scale bar: 1 µm. (**E**) Young’s Modulus obtained by nanoindentation on pure PDMS (blue), non-irradiated PDMS/Au NRs films (yellow), irradiated PDMS/Au NRs films (PDMS/Au NRs NIR, red), and post irradiated films (PDMS/Au NRs NIR OFF, black). Films were prepared at 2000 rpm for 1 min with 7.5 wt.% of Au NRs. One-way analysis of variance (ANOVA), post hoc Tukey’s test, **** *p* ≤ 0.0001.

**Figure 2 biomedicines-11-00030-f002:**
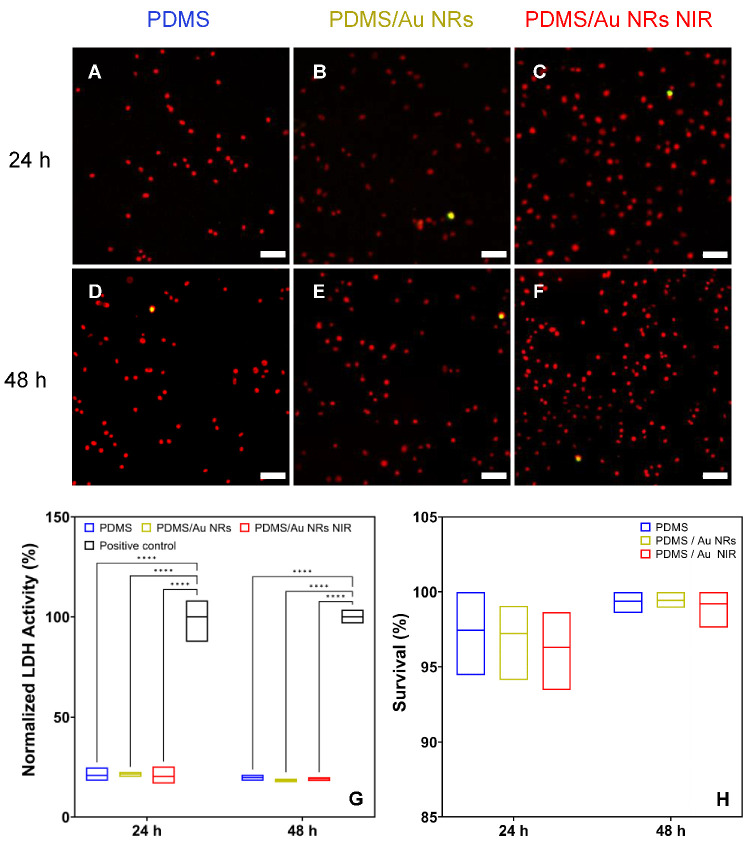
(**A**–**F**) cLSM images of fibroblast plasma membranes cultured on the PDMS films and stained with a commercial LIVE/DEAD assay showing live cells in red and cells with a compromised plasma membrane in green. Scale bar 80 µm. (**G**) LDH assay performed at 24 and 48 h for three independent biological replicates. Triton X-100 was used as a positive control. (**H**) Cell survival rate cultured on the control and irradiated films at 24 and 48 h of cultivation. Two-way ANOVA, post hoc Tukey’s test, **** *p* ≤ 0.0001.

**Figure 3 biomedicines-11-00030-f003:**
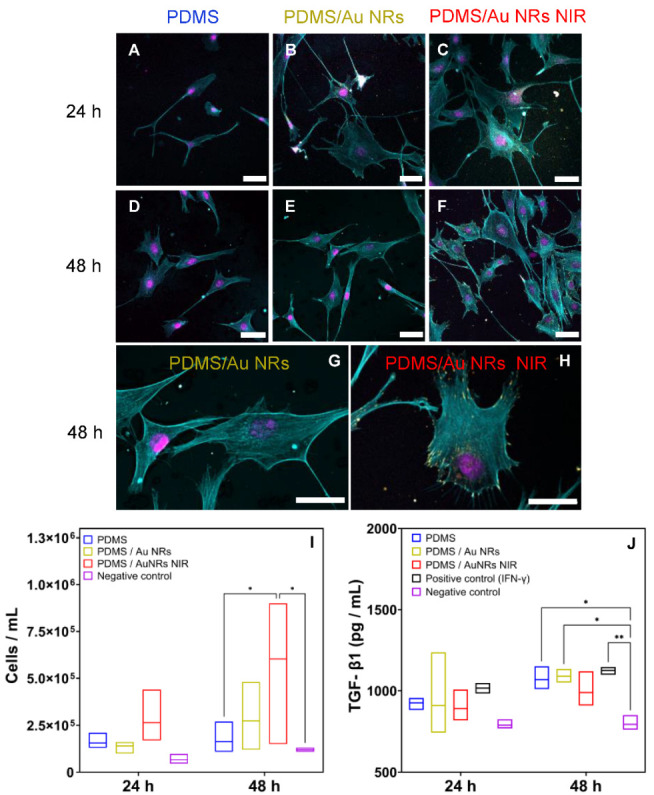
(**A**–**F**) cLSM images of NIH/3T3 fibroblast cultured on the films for 24 and 48 h. Cell nuclei (magenta), f-actins (cyan), and paxillin (yellow) were stained, respectively. Scale bar 40 μm. (**G**,**H**) Zoomed images of the fibroblasts cultured on non-irradiated PDMS/Au NRs films and PDMS/Au NRs NIR films at 48 h of cultivation. Scale bar 40 μm. (**I**) Number of NIH/3T3 by automatic counting using an EVE^TM^ counter (*n* = 3) after 24 and 48 h. (**J**) Secretion of TGF-β1 after 24 and 48 h (19000 cells/cm^2^). IFN-γ (1 μg/mL) was used as a positive control. Glass was employed as the negative control. Two-way ANOVA, post hoc Tukey’s test, * *p* ≤ 0.05, ** *p* ≤ 0.01.

## Data Availability

All raw data used to create the presented figures and tables can be found at https://doi.org/10.5281/zenodo.7292903 (accessed on 16 December 2022).

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
