# Peer review of "A Near-Infrared Mechanically Switchable Elastomeric Film as a Dynamic Cell Culture Substrate"

_biomedicines, 2022, doi:10.3390/biomedicines11010030_

Round 1

Reviewer 1 Report

This is a nice paper describing the production and characterization of a substrate for cell culture, modified by incorporation of gold nano rods to make the mechanical properties tunable by near-infrared light. 
The paper is well written, the methods thoroughly described and the results interesting. 

I only have  minor requests for the authors.
Minor
1. can you please add some possible applications of your films in tissue enginneering and also cite the tissues that respond dynamically to variations of stiffness 
2. can you please add a point in discussion on the effect (if significant or not) of the choice of the Poisson coefficient for the sample. Did you perform a sensitivity analysis ? If not, why ? page 4, line 193
3. the reported average Young's modulus of roughly 13MPa is a little high even for cartilage, and would be very high for softer tissues. Is there a way to modulate the Young's modulus to make it lower, in the order of 1MPa, or kPa even ? If yes, how ? Can you add a discussion paragraph on this point please ?

Round 2

Reviewer 2 Report

I thank the authors for their clarification regarding the NIR irradiation. It is clear with the current edits that irradiation occurs for the entire duration of cell culture (up to 24 hours). The edits/changes have addressed all of my concerns.